SOFTWARE

# resevol: An R package for spatially explicit models of pesticide resistance given evolving pest genomes

**A. Bradley Duthie**[1]*, **Rosie Mangan**[1], **C. Rose McKeon**[1], **Matthew C. Tinsley**[1], **Luc F. Bussière**[2,3]

**1** Biological and Environmental Sciences, University of Stirling, Stirling, United Kingdom, **2** Biological and Environmental Sciences and Gothenburg Global Biodiversity Centre, The University of Gothenburg, Gothenburg, Sweden, **3** Gothenburgh Global Biodiversity Centre, Gothenburg, Sweden

* alexander.duthie@stir.ac.uk

**Data Availability Statement:** The resevol R package can be downloaded from CRAN (https://cran.r-project.org/package=resevol) or GitHub (https://bradduthie.github.io/resevol/). The package is open source under GNU Public License.

**Funding:** This software was developed as part of the project for Enhancing Diversity to Overcome Resistance Evolution (ENDORSE) led by Luc Bussière, Ricardo Polanczyk, and Matthew Tinsley (Co-Investigators: Nils Bunnefeld, Yelitza Colmenarez, Natália Corniani, Renata de Lima, Brad

## Abstract

The evolution of pesticide resistance is a widespread problem with potentially severe consequences for global food security. We introduce the resevol R package, which simulates individual-based models of pests with evolving genomes that produce complex, polygenic, and covarying traits affecting pest life history and pesticide resistance. Simulations are modelled on a spatially-explicit and highly customisable landscape in which crop and pesticide application and rotation can vary, making the package a highly flexible tool for both general and tactical models of pest management and resistance evolution. We present the key features of the resevol package and demonstrate its use for a simple example simulating pests with two covarying traits. The resevol R package is open source under GNU Public License. All source code and documentation are available on GitHub.

## Introduction

Insect resistance to pesticides is a wicked and widespread problem [1]. Predicting, identifying, and ultimately delaying the evolution of insecticide resistance is therefore critical for ensuring sustainable global food security [2, 3]. To manage pesticide resistance, various strategies have been proposed to disrupt or weaken selection. These strategies often focus on varying the location and timing of pesticide application, and therefore varying the strength of selection for pesticide resistance, so that pest susceptibility to one or more pesticides is maintained [4–8, 10]. Pesticide resistance management has mostly been associated with the effect of single genes. In such cases, resistance alleles have binary effects on phenotype, enabling resistant phenotypes to arise from genetic changes at single or a small number of loci [11–13]. Nevertheless, resistance attributable to polygenic effects is also well-established [14–19]. And the relevance of polygenic resistance is likely to further increase given a rising interest in biological controls for sustainable crop protection [e.g., 18–20]. Developing strategies to maximise the efficacy of these tools is critical, and such strategies should be well-informed by predictions made from detailed, quantitative genetic models. Here we introduce

Duthie, Steve Edgington, Leonardo Fraceto, Belinda Luke, and Rosie Mangan). The ENDORSE project is a joint Newton funded international partnership between the Biotechnology and Biological Sciences Research Council (BBSRC) in the UK and the São Paulo Research Foundation (FAPESP) in Brazil under BBSRC award reference BB/S018956/1 and FAPESP award reference 2018/21089-3. ENDORSE is a partnership among Universidade Estadual Paulista (UNESP), the University of Stirling (UoS), and the Centre for Agricultural and Biosciences International (CABI). The funders had no role in study design, data collection and analysis, decision to publish, or preparation of the manuscript.

**Competing interests:** The authors have declared that no competing interests exist.

the resevol R package as a tool for building individual-based models and simulating pest management [21].

The resevol package applies individual-based modelling and a quantitative genetics approach to simulate the evolution of a pest population on a changing landscape. Multiple traits determine the overall fitness of any pest genotype. For example, while alleles conferring resistance to a particular pesticide enhance fitness in the presence of that pesticide, they could also impose reproductive fitness costs. Such trade-offs can be quantified by the genetic covariance between traits. A focal goal of the resevol package is to model traits with a pre-specified, but potentially evolving, genetic covariance structure. To achieve this goal, each individual has a genome with $L$ loci that underlie a set of $T$ potentially evolving traits. Pleiotropic loci can vary in their effects on the direction and magnitude of polygenic traits, causing population-wide trait covariance to arise mechanistically from the underlying genetic architecture of individuals. To achieve this, two separate steps are necessary. First, an evolutionary algorithm is used to find a network of internal nodes that map standard random normal loci values to covarying traits. Values used to map loci to traits are incorporated into individual genomes. Second, a population of asexual or sexual individuals is initialised and simulated on a spatially explicit landscape separated into distinct units (e.g., farms). Land units can apply one of up to 10 pesticides, and one of up to 10 landscape types at a given time (for simplicity, here we interpret landscape types to be crop types). Pesticides and crops rotate independently within farms over time in a pre-specified way. The resevol package can thereby model complex and evolving agricultural pest traits over realistic landscapes that undergo different pesticide use and crop regimes.

## Design and implementation

### Covarying pest quantitative traits

The first step of simulation is building individual genomes. This step is separate because it is computationally intensive, and genomes that are built might need to be inspected and stored. High computation time is due to the mechanistic nature of how genomes and covarying traits are modelled. Instead of imposing a trait covariance structure directly, an evolutionary algorithm is used to find a network that maps standard random normal values (loci) to covarying values (traits). This is useful because it allows genomes to model potentially evolving physiological constraints, and trade-offs among traits, from the bottom up. Since multiple networks can potentially map loci to the same trait covariance structure, it is possible to replicate evolution with different randomly generated genetic architectures. This approach to modelling individual genomes and traits thereby increases the complexity of questions that can be addressed for simulating evolution in agricultural pests.

Conceptually, the relationship between individual genotypes and traits is defined by a network connecting loci to traits through a set of hidden internal nodes (Fig 1). Values for loci are randomly drawn from a standard normal distribution, $\mathcal{N}(0, 1)$. Links between loci, internal nodes, and traits, can take any real value and are represented by black arrows in Fig 1. Traits can take any real value, and are calculated as the summed effects of all preceding nodes (i.e., the blue squares immediately to the left of the traits in Fig 1). Mathematically, loci are represented by a row vector of length $L$. Effects of loci on the first layer of internal nodes (black arrows emanating from loci in Fig 1) are represented by an $L \times T$ matrix, and transitions between internal nodes, and between the last set of internal nodes and the final traits, are

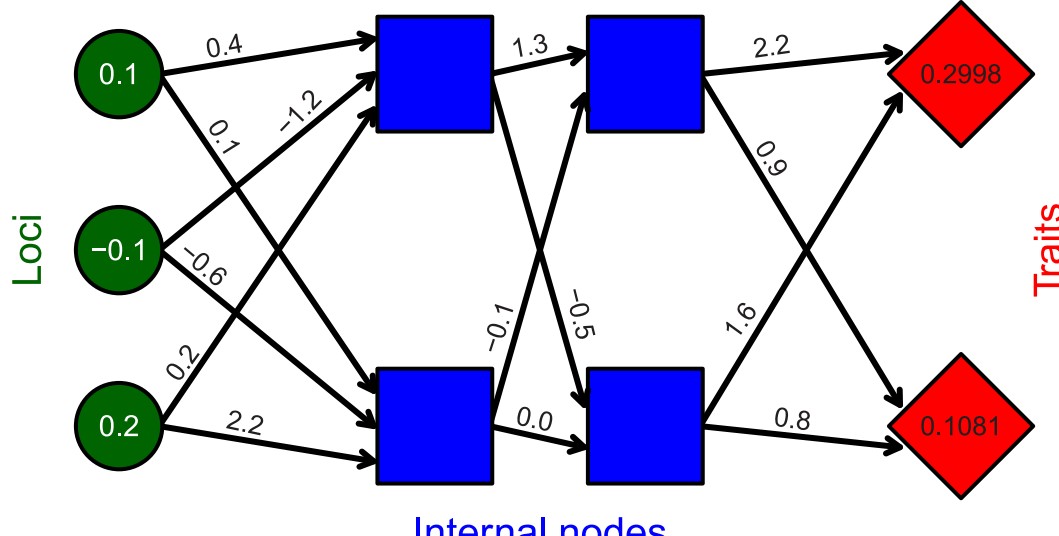

**Fig 1. Example resevol network.** Example network mapping loci (green circles) to traits (red diamonds) through an intermediate set of hidden layers (blue squares) in the `mine_gmatrix` function. Individual genomes in the resevol R package consist of standard random normal values for loci, real values for black arrows linking nodes, and real values for traits. Values shown for loci and arrows are an example for illustration.

represented by $T \times T$ matrices. For the example in Fig 1,

$$\begin{pmatrix} 0.1, & -0.1, & 0.2 \end{pmatrix} \begin{pmatrix} 0.4, & 0.1 \\ -1.2, & -0.6 \\ 0.2, & 2.2 \end{pmatrix} \begin{pmatrix} 1.3, & -0.5 \\ -0.1 & 0.0 \end{pmatrix} \begin{pmatrix} 2.2, & 0.9 \\ 1.6 & 0.8 \end{pmatrix} = \begin{pmatrix} 0.2998, & 0.1081 \end{pmatrix}.$$

Values mapping loci to traits become part of an individual's genome, so the genome for the individual represented by Fig 1 is stored in the model as shown below:

```
0.1, -0.1, -0.2, 0.4, 0.1, -1.2, -0.6, 0.2, 2.2, 1.3, -0.5,
-0.1, 0.0, 2.2, 0.9, 1.6, 0.8
```

Individuals with different loci can therefore have different covarying traits that are constrained by the network structure encoded in each genome. Individuals can be haploid (as in Fig 1) or diploid (in which case, allele values are summed at homologous loci).

An evolutionary algorithm is used to find appropriate values that produce covarying traits from loci (see S1 Text for details). Evolutionary algorithms are heuristic tools that can simulate adaptive evolution to find solutions for a broad range of problems [22, 23]. In the resevol package, the `mine_gmatrix` function runs an evolutionary algorithm and requires the argument `gmatrix`, which specifies the desired trait covariance matrix. The function initialises a population of `npsize` separate, and potentially unique, networks (i.e., `npsize` copies of a network like the one shown in Fig 1), and this population evolves until some maximum iteration (`max_gen`) or minimum expected network stress (`term_cri`) is met. In a single iteration of the algorithm, values mutate and crossover occurs between networks (i.e., some of the `npsize` networks swap values with some probability). Next, trait covariances produced for each network are estimated by initalising `indivs` individuals with loci sampled from a standard normal distribution. Network stress is calculated as the logged mean squared deviation between estimated covariances and those in `gmatrix`. Tournament selection [22] is then

used to determine the networks for the next iteration of the algorithm. Networks with the lowest estimated stress have the highest fitness, so these networks are disproportionately represented in the next iteration. Throughout the evolutionary algorithm, the lowest stress network is saved and returned upon network termination. The robustness of this network's stress to sets of individuals with different loci values can be tested using the `stress_test` function.

An example run of `mine_gmatrix` with the same number of loci and internal nodes as in Fig 1 is shown below for traits that have an intended covariance of -0.4:

```
trait_covs <- matrix(data = c(1, -0.4, -0.4, 1), nrow = 2, ncol = 2);
new_network <- mine_gmatrix(loci = 3, layers = 2, gmatrix = trait_covs,
                            max_gen = 1000, term_cri = -6.0,
                            prnt_out = FALSE);
```

The code above found a genome that produced the following expected trait covariances:

```
##            [,1]       [,2]
## [1,]  1.0008181 -0.3925799
## [2,] -0.3925799  1.0271340
```

The mean deviation between elements of the above matrix and the identity matrix provided by `trait_covs` is $2.1176024 \times 10^{-4}$. Lower values of `term_cri` and higher values of `max_gen` will result in a lower stress, but this will require more computation time, especially if the number of traits is high. Similarly, higher values of `indivs` will result in more accurate estimations of true stress, but this also requires more computation time. Additional arguments to `mine_gmatrix` can also be used to improve the performance of the evolutionary algorithm (see S1 Text).

## Simulating landscape-level pesticide resistance

The full output of `mine_gmatrix` is passed to the `run_farm_sim` function, which initialises and simulates an evolving population of pests on a changing landscape for any natural number of time steps (`time_steps`). In this section, we explain the landscape, pest ecology, and evolving pest traits.

**Landscape.**   Landscapes are spatially explicit and initialised in one of two ways. First, a landscape can be built from the arguments `xdim`, `ydim`, and `farms`. These arguments specify the cell dimensions and number of farms on the landscape. Contiguous rectangular farms of roughly equal size are generated on the landscape using a splitline algorithm. Second, a custom landscape can be input using the `terrain` argument, which takes a matrix with elements that include integers 1 to `farms`. Each value defines a unique farm, but values do not need to be contiguous. These 'farms' could even model non-farmland (e.g., water, roads), if pesticides and crops on them are invisible to pests (see S2 Text), or they could more broadly be interpreted as heterogenous landscape properties across an arbitrary scale [e.g., 24]. This `terrain` customisation therefore allows for a high degree of landscape detail, and offers the potential for modelling real-world landscapes from raster images [e.g., 25]. Edge effects are set using the `land_edge` argument. Edge options include "`leaky`" (pests disappear from the landscape), "`reflect`" (pests bounce off of the edge), "`sticky`" (pests stick to the edge), or "`torus`" (pests that move off of one edge return on the opposite side of the landscape).

Each farm can hold one pesticide and one crop type in any time step. The `crop_init` and `pesticide_init` arguments initialise one of `crop_number` crops and one of `pesticide_number` pesticides for each farm, respectively (maximum of 10 each).

Initialisation can be random for each farm with equal probability, or it can be set using a vector of length `farms` in which vector elements define the initialised crop or pesticide number. After initialisation, crops and pesticides rotate once every `crop_rotation_time` and `pesticide_rotation_time` time steps, respectively. The arguments `crop_rotation_type` and `pesticide_rotation_type` specify how crops and pesticides are rotated, respectively. Each of these rotation type arguments can take either an integer value from 1–3, or a matrix. Integer values specify (1) no rotation, (2) random transition from one type to another, or (3) cycling through each available crop or pesticide in numeric order. Square matrices can be used to define the probability that a given crop or pesticide in row *i* transitions to that in column *j*. Hence, any possible Markov chain can be used to model transition between crop or pesticide types on farms, potentially integrating real crop use or pesticide use patterns [26]. Upon rotation, crop and pesticide values are reset on each farm.

Values for `crop_per_cell` and `pesticide_per_cell` determine the quantity of crops and pesticides initialised per cell upon crop or pesticide rotation, respectively. Between crop rotations, crop values can increase each time step by a proportion or increment `crop_growth`, which depends on `crop_growth_type`. These arguments can be used to model crop growth over a season.

**Pest ecology.**   Individual pests can be modelled to have several reproductive systems and life histories. Pest reproductive system can be specified using the `repro` argument, which accepts "`asexual`" (haploid), "`sexual`" (monoecious), and "`biparental`" (dioecious). For "`sexual`" pests, the `selfing` argument specifies if self-fertilisation is (`TRUE`) or is not (`FALSE`) allowed. At the start of a simulation, pests are initialised in a random location. Initialised pests are of age zero if `rand_age = FALSE` or a random age from zero to `max_age` if `rand_age = TRUE`. Following initialisation, a single time step proceeds with landscape change (see above), pest aging and metabolism, feeding, pesticide consumption, movement, reproduction, mortality, and immigration (Fig 2). Feeding, pesticide consumption, movement, and reproduction all depend on pest age. Pests feed and consume pesticide from ages `min_age_feed` to

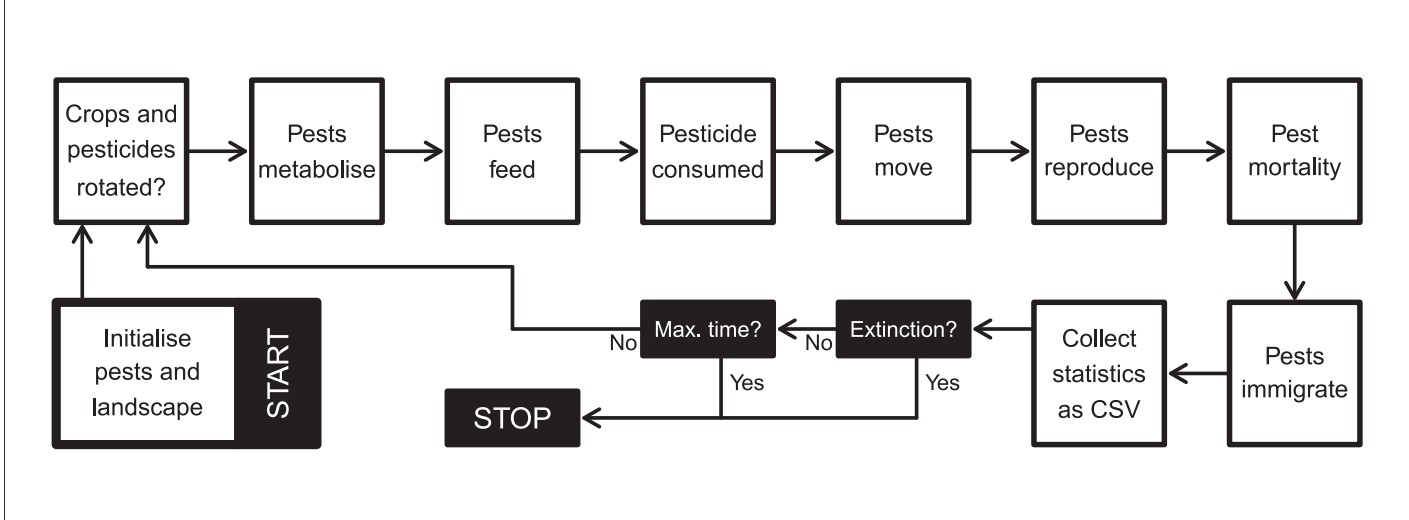

**Fig 2. Overview of resevol events.** Overview of simulated events in the resevol R package. Note that metabolism, feeding, pesticide consumption, movement, and reproduction are all subject to a minimum and maximum pest age. Consequently, simulation order might not reflect the order of events from the perspective of a focal pest (e.g., pests might move from ages 1–2, but only feed from ages 2–4). Crops and pesticides are also not necessarily rotated in each time step (see Landscape). Statistics collected within a time step are printed to a CSV file.

max_age_feed, move from ages min_age_move to max_age_move, and reproduce from ages min_age_reproduce to max_age_reproduce (all inclusive). Food accumulated is lost during aging if baseline_metabolism > 0 and pest age is within min_age_metabolism and max_age_metabolism. The option to set minimum and maximum ages for events makes it possible to model pests with much different life histories [e.g., 9].

In each time step, pests feed in a random order, consuming the crop on their landscape cell. Each pest consumes an amount of crop as specified by the food_consume argument, which takes a vector with as many elements as their are crops (i.e., if crop_number = 2, then food_consume has two elements, the first and second defining consumption of crops 1 and 2, respectively). If crop amount on the landscape cell exceeds pest consumption ability, then pests consume their maximum amount, and this amount is removed from the landscape cell. If crop amount is less than pest consumption ability, then pests consume whatever crop is left and crop amount is reduced to zero. Pesticide consumption works identically to crop consumption, except that the amount of pesticide on a landscape cell is not decreased. Hence, each pest can potentially feed and be affected by the pesticide of their focal landscape cell. Pests simply consume an amount of pesticide as specified by the pesticide_consume argument, which also takes a vector with as many elements as their are pesticides. Pesticide consumption affects pest survival and reproduction using pesticide_tolerated_surv and pesticide_tolerated_repr arguments.

After interacting with their landscape cell, pests can move. Each pest visits a number of cells during movement, as is specified by the parameter movement_bouts. Individual movement bouts occur in a random order across pests. During a movement bout, a pest can travel to any cell within a value defined by movement_distance from their current location, which could include their current location (i.e., moving zero distance). Upon arrival to a cell, a pest can feed if they are of an appropriate feeding age and feed_while_moving = TRUE. The pest also consumes pesticide if they are of the appropriate age and pesticide_while_moving = TRUE. Having pests feed and consume pesticide in a random order while moving among landscape cells can model a population competing for food and encountering pesticides on a shorter time scale than an individual time step.

After all pests finish moving, pests reproduce. Offspring production is possible for asexual, monoecious, or female pests. Pest expected offspring number is defined by a fixed parameter if reproduction_type = "lambda", or is calculated from the amount of food consumed if reproduction_type = "food_based". The former requires specifying lambda_value, which becomes the rate parameter for sampling offspring number from a Poisson distribution. The latter requires specifying a real value for food_needed_repr, which is the amount of food needed to produce one offspring. For food-based reproduction, the total amount of food consumed is divided by food_needed_repr, then floored to determine offspring number. Sexual reproduction requires a mate of reproductive age that is either monoecious or male, and within range of the reproducing focal pest (potential including the focal pest, if selfing = TRUE). A potential mate is within range if it is within an integer number of cells from the focal pest, as defined by mating_distance (e.g., if mating_distance = 0, then mates must share a landscape cell). All available potential mates sire offspring with equal probability, and reproducing pests are assumed to mate multiply (i.e., paternity is a fair raffle for all offspring). If a carrying capacity at birth is set (K_on_birth > 0) and total offspring number in the population exceeds this capacity, then offspring are removed at random until they are within carrying capacity. A real value immigration_rate specifies the rate parameter for Poisson random sampling of the number of immigrants added to the population at the end of a time step. Immigrants are initialised

in the same way as pests are at the start of the simulation. Hence, any spatial structure or evolution that occurs during the simulation does not affect immigrant locations, genomes, or traits.

**Pest evolution.** Pest genomes evolve in a complex and highly mechanistic way. Offspring inherit genome values from their parent(s) with the possibility for mutation and recombination; offspring traits are then calculated from their newly initialised genomes. For asexually and sexually reproducing pests, genomes are haploid and diploid, respectively. Asexually reproducing pests receive the full genomes of their parent, while sexually reproducing pests receive half of their alleles from each parent. Each diploid parent contributes one half of their genome, effectively modelling a genome with a single chromosome. Crossover occurs at each position of the genome with a probability of `crossover_pr`. When a crossover event occurs, alleles are swapped between the two chromosomes, so complete recombination is also possible if `crossover_pr = 0.5`. For both haploids and diploids, each genome value then mutates independently with a probability of `mutation_pr`, which can be set to any real number from 0–1. If a genome value mutates, then a new value is randomly sampled from a standard normal distribution. If `mutation_type = 0`, then this new value replaces the old value, and if `mutation_type = 1`, then the new value is added to the old value. After mutation, genome values are used to calculate trait values.

Evolution of the genetic architecture linking loci to traits can be constrained by disabling mutation in genome values linking loci, internal nodes, and traits (i.e., 'network values' represented by arrows in Fig 1). While mutation at loci (green circles in Fig 1) is always possible as long as `mutation_pr > 0`, the number of intermediary layers for which network values can mutate is constrained by `net_mu_layers`. If `net_mu_layers = 0`, then no network values can mutate, but higher integer values cause mutation to occur at network value layers from loci to traits (if `net_mu_dir = 1`) or traits to loci (if `net_mu_dir = 0`). For example, if `net_mu_layers = 2` and `net_mu_dir = 1`, then the network values linking loci to the first internal node, and the first internal node to the second, can mutate (i.e., first two columns of arrows in 1, but not those linking the second internal node to traits). This allows pest traits to evolve with varying degrees of constraint on the covariance between traits. Low `net_mu_layers` values model strong genetic constraints, while high values model high evolvability of trait covariances.

Finally, evolving and covarying traits can be used in place of fixed parameters described in pest ecology. This is done by substituting "Tj" as an argument input in place of a numeric value, where j represents the trait number. For example, the argument `move_distance = "T1"` will make Trait 1 the movement distance for individuals. The argument `food_consume = c ("T2", "T3")` will set Traits 2 and 3 to define the amount of food of types 1 and 2 that can be consumed by a pest, respectively. Up to 10 of the following parameters can be replaced with evolving traits: `move_distance`, `food_needed_surv`, `pesticide_tolerated_surv`, `food_needed_repr`, `pesticide_tolerated_repr`, `mating_distance`, `lambda_value`, `movement_bouts`, `metabolism`, `food_consume`, and `pesticide_consume`. Mean values of traits can be set using the argument `trait_means`, which accepts a vector of the same length as the number of evolving traits such that indices correspond to trait numbers. The resevol package can thereby simulate agricultural pests with complex and co-evolving traits, and potentially evolving trait covariances, under a range of possible pest life histories.

## Simulation output

Simulation output is typically large, so output is printed in two CSV files, both of which are created in the working directory. The first file "population_data.csv" prints population level

data over time, including population size, mean age, sex ratio, mean food and pesticide consumed of each type, mortality rate, and mean trait values. The second file "individuals.csv" prints all information, including full genomes and traits (columns), for every individual (rows) in the population. The printing of individual level data is disabled by default. It can be turned on for all time steps by setting `print_inds = TRUE`, but this should be done with caution because it can create extremely large files. Instead, individual level data can be printed for only the final time step by setting `print_last = TRUE`. Output produced by `run_farm_sim` is a list of two elements, which includes a vector of parameter values used in the simulation and the final state of the landscape as an array.

## Results

In resevol v0.3, the `run_farm_sim` function includes 68 arguments, which specify a wide range of possible simulation conditions affecting landscape and pest characteristics. These arguments are explained in the package documentation, and in S2 Text, which demonstrates an advanced case study with a custom landscape and complex pest genomes and life history. Here we focus on a simple simulation with asexually reproducing pests that have three loci and two traits (Fig 1). We use the the genome generated in `new_network` from Section 1.1, in which traits 1 and 2 have variances of 1.0008181 and 1.027134, respectively, with a covariance of -0.3925799.

Traits 1 and 2 will define the realised rate of uptake of the two separate pesticides, so we model a system in which there is a potential trade-off for pesticide susceptibility. We use a simple $64 \times 64$ cell landscape with nine farms. Each farm grows the same crop and uses one of two randomly intialised pesticides, which are rotated every 16 time steps. Hence, we can conceptualise 16 time steps as a single growing season.

In our example, each cell produces four crop units, all of which can potentially be consumed. Pests consume up to one unit of crop on their landscape cell per time step. Pest survival and reproduction is food-based, and pests need to consume one unit of crop by age two to survive and reproduce. Pests initialised at the start of the simulation are randomly assigned an age from 0–4 with equal probability. Pests have a maximum age of four, and they feed in ages 0–2, move up to two cells in ages 3–4, and reproduce in age four. Pests can uptake (i.e., 'consume') pesticides in ages 0–2, and if they consume any pesticide, then they will die. In each time step, a mean of 10 immigrants arrive. We simulate 160 time steps (10 growing seasons) using the function below:

```
sim <- run_farm_sim(mine_output = new_network, repro = "asexual",
                    pesticide_number = 2, pesticide_init = "random",
                    pesticide_consume = c("T1", "T2"), farms = 9,
                    pesticide_rotation_time = 16,
                    pesticide_rotation_type = 3,
                    pesticide_tolerated_surv = 0,
                    pesticide_per_cell = 1,
                    crop_rotation_time = 16, crop_number = 1,
                    crop_per_cell = 4, food_consume = 1,
                    reproduction_type = "food_based",
                    food_needed_surv = 1, food_needed_repr = 1,
                    max_age = 4, min_age_feed = 0, max_age_feed = 2,
                    min_age_move = 3, max_age_move = 4,
                    min_age_reproduce = 4, print_gens = FALSE,
```

```
                                max_age_reproduce = 4, age_pesticide_threshold = 2,
                                rand_age = TRUE, move_distance = 2,
                                immigration_rate = 10, time_steps = 160,
                                print_last = TRUE, xdim = 64, ydim = 64,
                                trait_means = c(0.1, 0.1), land_edge = "torus");
```

Any arguments to `run_farm_sim` not included above are set to default values. Output files can be used to plot ecological and evolutionary dynamics of pests. Fig 3a shows the pest population change both across and within seasons. Due to the specific parameter values chosen, clear patterns reflecting pest cohorts emerge. At the start of a season when the most crop is available, pests eat and abundance increases. As less crop remains in a season, pest abundance decreases. Mean food consumed varies over the course of a season caused by the varying frequency of pests in different life history stages (Fig 3c). In the first 50 time steps, there is some consumption of pesticides 1 and 2 (Fig 3b), but both traits rapidly evolve to negative values making both pesticides ineffective (Fig 3d). In this case, the trade-off in pesticide consumption is not strong enough to maintain susceptibility to either pesticide in the population. Fig 4 shows the location of pests on the landscape in the last time of the simulation (left panel), and how different pesticides are currently being applied on the landscape (right panel). Because the number of crops and pesticides, the number and nature of traits, and the size of pest genomes all vary depending on simulation parameters, no plotting functions are introduced in the

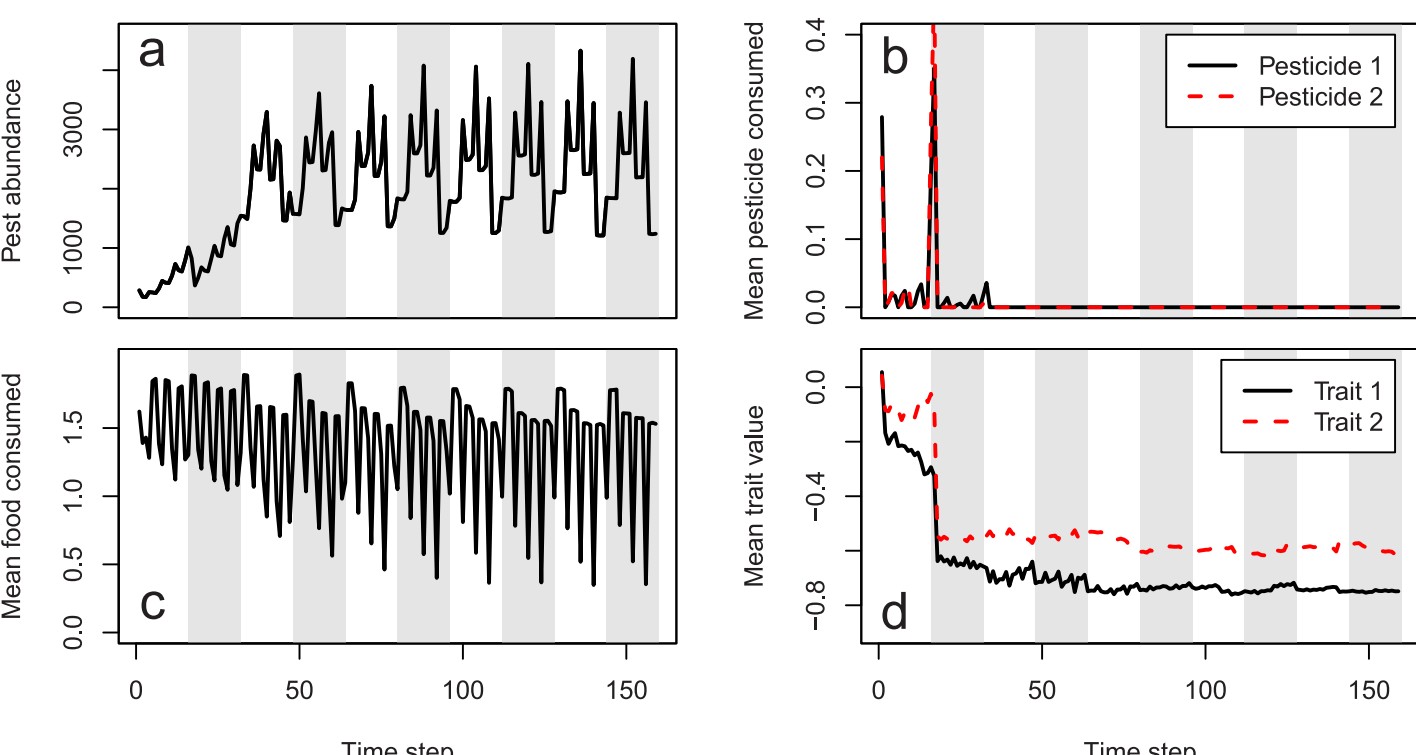

**Fig 3. Example resevol population dynamics.** Agricultural pest ecological and evolutionary dynamics over 160 time steps from an individual-based simulation using the resevol R package. Panels show (a) pesticide abundance change, (b) mean realised amount of pesticides 1 and 2 uptaken per pest, (c) mean food consumed per pest, and (d) mean value of evolving traits 1 and 2 underlying pest uptake over time. Note that only pests with positive values for Traits 1 or 2 can uptake Pesticide 1 or 2, respectively. Pests with negative trait values will be unaffected by corresponding pesticides (i.e., pesticide consumption is a threshold trait), hence the difference between realised pesticide consumption (b) and the traits underlying it (d). White and grey vertical stripes indicate seasons of a single crop and pesticide application.

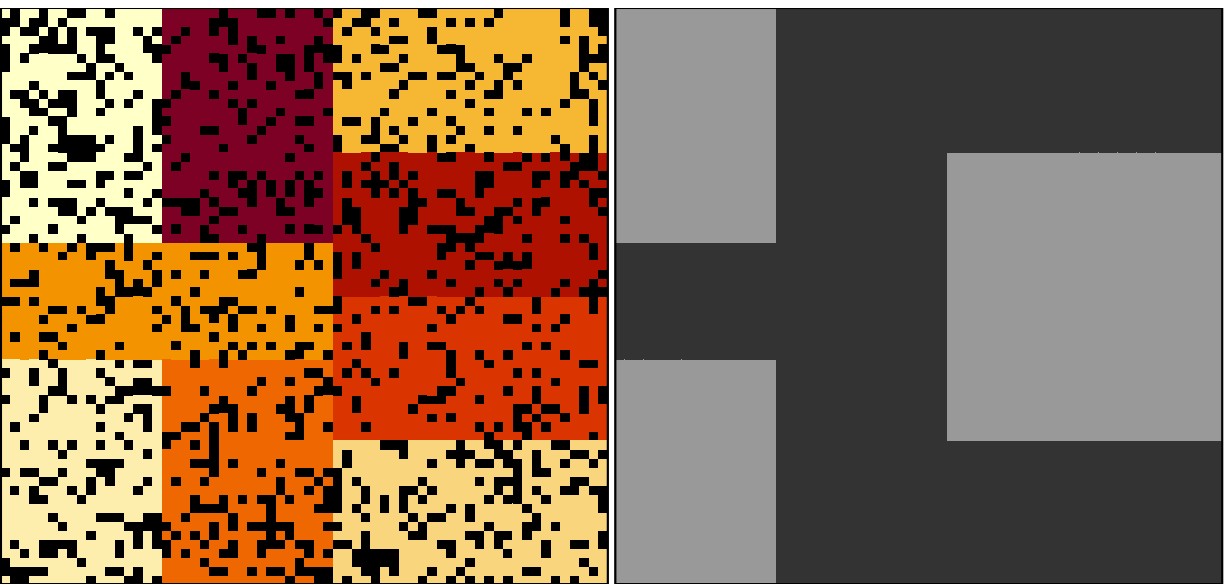

**Fig 4. Pest spatial distributions in resevol simulation.** Locations of pests (black) across a landscape that includes nine farms (coloured blocks) in the last time step of a simulation using the resevol R package (left panel). The right panel shows which farms apply pesticide 1 (dark grey) and 2 (light grey).

resevol package. Instead, methods and code for producing plots such as those in Figs 3 and 4 are explained in S2 Text.

## Availability and future directions

Insecticide resistance to pesticides is a widespread problem that affects global food security [3, 27, 28]. Many models have been developed to investigate the evolution of resistance under different ecological and evolutionary conditions [e.g., 4, 5, 7–10, 17, 29, 30]. The resevol R package makes it possible to rapidly develop and simulate myriad individual-based models of resistance evolution. It also introduces a novel approach to modelling complex pest genetic architecture, using an evolutionary algorithm to generate haploid or diploid loci that map to pest traits with pre-specified covariances (Fig 1; S1 Text). Agricultural landscapes and pest life histories are highly customisable, allowing targetted models that can simulate specific real-world case studies (S2 Text). The breadth of possible models that can be simulated with resevol also makes it a useful tool for developing theory on pest management, and even more generally on the evolution and ecology of individuals with complex traits on a heterogeneous landscape.

## Supporting information

**S1 Text. The evolutionary algorithm.** The evolutionary algorithm of the resevol R package, including key data structures used, a general overview of the evolutionary algorithm, and details concerning haploid and diploid individuals.
(PDF)

**S2 Text. Advanced techniques.** Advanced techniques for using the resevol R package, including initialising pest genomes and running simulations.
(PDF)

## Author Contributions

**Conceptualization:** A. Bradley Duthie, Rosie Mangan, C. Rose McKeon, Matthew C. Tinsley, Luc F. Bussière.

**Funding acquisition:** A. Bradley Duthie, Matthew C. Tinsley, Luc F. Bussière.

**Methodology:** A. Bradley Duthie, Luc F. Bussière.

**Project administration:** Matthew C. Tinsley, Luc F. Bussière.

**Software:** A. Bradley Duthie, C. Rose McKeon.

**Supervision:** Luc F. Bussière.

**Writing – original draft:** A. Bradley Duthie, Rosie Mangan.

**Writing – review & editing:** A. Bradley Duthie, Rosie Mangan, C. Rose McKeon, Matthew C. Tinsley, Luc F. Bussière.

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
