## [Decision Letter · Decision Letter 0]

15 Nov 2023

Dear Dr Duthie,

We are pleased to inform you that your manuscript 'resevol: an R package for spatially explicit models of pesticide resistance given evolving pest genomes' has been provisionally accepted for publication in PLOS Computational Biology.

Best regards,

Robert Noble, Ph.D.

Guest Editor

PLOS Computational Biology

Zhaolei Zhang

Section Editor

PLOS Computational Biology

Although neither author recommends any revisions, you may choose to act on some of their comments regarding the manuscript and the code. Please note that "PLOS ONE" in Reviewer 2's report should read "PLOS Computational Biology".

Reviewer's Responses to Questions

**Comments to the Authors:**

Reviewer #1: The algorithms in this package seem well-validated and the code includes the ability to simulate resistance evolution given specific scenarios and landscapes. In actual analysis situations, there remains the issue of how to actually measure genetic parameters such as covariance between genes and environmental parameters such as carrying capacity. However, determining plausible parameter sizes is a separate issue from the completeness of the program itself, and should be explained in a separate article.

Reviewer #2: The paper presents a spatially-explicit evolutionary modeling framework to investigate insect resistance to pesticides. For food security and biological conservation, the subject is highly critical, and it has been a pleasure to review this new R package.

I recommend the publication of the article in PLOS ONE.

I have a minor comment: The package architecture is very good. I understood where the elements are and how the scripts connect to each other. However, given the complexity of the model, the functions are sometimes quite lengthy to read. It could have been interesting to factorize the code (create smaller functions that you then call). This observation seems important both for internal code, at least for the maintainability of the package over time (since the user is not supposed to look at it), and also for functions intended for the user. For instance, the 'run_farm_sim' function has many elements. It might be wise to break it down into steps to more easily define the model to simulate. You did it for the ‘mine_gmatrix’. For example, as you presented in the article, in three steps: defining the landscape, then the phenology of the pest, and finally the mechanisms of evolution, and also the pesticide part. The underlying idea of making things modular is that as the package grows, each of the elements (landscape, pesticide, population, and genetics) will become more complex, and you may end up with a very complex system.

**Have the authors made all data and (if applicable) computational code underlying the findings in their manuscript fully available?**

Reviewer #1: Yes

Reviewer #2: Yes

PLOS authors have the option to publish the peer review history of their article (what does this mean?). If published, this will include your full peer review and any attached files.

Reviewer #1: No

Reviewer #2: No

---

## [Editor Report · Acceptance letter]

27 Nov 2023

PCOMPBIOL-D-23-01285 

resevol: an R package for spatially explicit models of pesticide resistance given evolving pest genomes

Dear Dr Duthie,

I am pleased to inform you that your manuscript has been formally accepted for publication in PLOS Computational Biology. Your manuscript is now with our production department and you will be notified of the publication date in due course.

With kind regards,

Zsofi Zombor
